# An Assessment of Factors Related to Ocean Literacy Based on Gender-Invariance Measurement

**DOI:** 10.3390/ijerph16193672

**Published:** 2019-09-30

**Authors:** Liang-Ting Tsai, Yen-Ling Lin, Cheng-Chieh Chang

**Affiliations:** 1Taiwan Marine Education Center, National Taiwan Ocean University, Keelung 20224, Taiwan; linyenling0619@gmail.com; 2Institute of Education & Center of Teacher Education, National Taiwan Ocean University, Keelung 20224, Taiwan

**Keywords:** ocean literacy, environmental education, parental education level, attitude toward ocean, measurement invariance, multiple-group SEM

## Abstract

This study sought to gain a more comprehensive understanding of how the factors of parental education level and student attitude toward the ocean influence the ocean literacy of students in Taiwan after establishing measurement invariance across genders. The analyzed data were collected from self-reported questionnaires filled out by students aged 16–18 years old. The students’ ocean literacy was used as the outcome variable, while parental education level and student attitude toward the ocean were employed as the independent variables. The effects of parental education level and student attitude toward the ocean on ocean literacy were estimated with a multi-group structural equation model. Of the final total of 945 valid respondents in this study, 58.1% were male and 41.9% were female. The results from the multiple-group analysis supported measurement invariance across the genders. After establishing gender invariance, it was further found that higher degrees of parental education level and student attitude toward the ocean were positively related to ocean literacy. A considerable contribution was detected between parental education level and ocean literacy that was indirectly related through student attitude toward the ocean in the female student.

## 1. Introduction

The relationships between students’ ocean literacy, parental educational level (PEL), and attitude toward the ocean (ATO) have not been well explored in previous research, especially across different groups and cultures. However, these relationships are quite important for learning about the ocean. As such, understanding these relationships can help us to understand the learning paradigm more clearly and provide different perspectives across cultures. It can also help us compare various educational practices and improve the learning progress of students with respect to ocean literacy. However, when research focuses on cross-group comparisons, such as comparisons across different genders, cultures, and nationalities, the instrument utilized needs to meet the requirement of measurement invariance across the groups. Nonetheless, numerous studies undertaking such cross-group comparisons have given little or no attention to this issue.

If a study does not ensure that measurement invariance has been established, then the possibility that some of the research participants may be more or less likely to give specific responses to the measure, which would in turn affect their scores on it, cannot be excluded [1]. Moreover, because this phenomenon is systematic, it will result in bias that can lead researchers to incorrect conclusions. For example, when there is actually no difference between two or more different groups, researchers may reach the opposite and incorrect conclusion that the groups have differences. Or, when there actually are differences between the different groups, the researchers may come to the incorrect conclusion that there are no differences. In short, if a survey study does not first prove the measurement invariance of the instrument that it utilizes, then the conclusion of the study may be inaccurate, which may, in turn, make the conclusion more difficult to explain.

In a study on ocean literacy, it is thus necessary to establish measurement invariance in order to ensure that false conclusions are not reached with respect to the relationships between ocean literacy, gender, and other variables. Relatedly, since many studies have reported that students exhibit gender differences in ocean literacy, the establishment of measurement invariance across genders is particularly important for exploring the relationships among various background variables, contextual factors, and ocean literacy. Thus, this study aimed to analyze how the factors of PEL and student ATO affect ocean literacy after establishing measurement invariance across genders among a sample of Taiwanese students aged 16–18 years old.

## 2. Literature Review

### 2.1. Gender and Ocean Literacy

A number of past studies have conducted investigations of how male and female students differ with respect to marine science knowledge, marine science misconceptions, and ocean literacy [2,3,4]. For example, a study of junior high students conducted by Chang [2] found that male and female students exhibited no significant differences in terms of ocean literacy knowledge. However, a study looking at students attending five senior high schools in Taiwan conducted by Lwo et al. [3] found that the marine science scores of the female students were significantly higher than those of the male students. Meanwhile, Steel et al. [4] conducted a survey of marine science knowledge among 1233 citizens over 18 years of age in the United States. The results indicated that the respondents living in coastal regions had more marine and coastal knowledge than the respondents living in non-coastal areas. Indeed, the coastal respondents had higher correct rates for all aspects of the ocean literacy quiz than the non-coastal respondents. The research also found that newspapers and the internet are likely to improve citizen knowledge of ocean-related issues, while television and radio have a negative effect. The study further found that males are significantly more accurate in self-assessing their marine knowledge and ocean quiz scores than females [4]. In addition, some studies comparing responses between the genders have found that women are more concerned about the issues facing the marine environment than men are [5,6].

### 2.2. Parental Education and Ocean Literacy

The relationship between socioeconomic factors and ocean literacy has not been widely discussed in the previous literature. However, the inference that socioeconomic factors (e.g., parental education level, parental income) have some impact on science performance or scientific literacy has been a major concern in related research studies [7,8]. In particular, the critical nature of the link between scientific achievement and parental education levels has been heavily emphasized by a number of past studies [1,7,9,10,11,12,13]. Parental education is considered to be one of the most stable aspects of socioeconomic status because it is usually established at an early age and tends to remain constant over time [1,14]. Kalender and Berberoglu [7] used parental education as an indicator of socioeconomic status and pointed to it as one of the important variables that affect students’ scientific achievement. Caldas and Bankston [15] also used parental education to represent family socioeconomic status and explored the influence of such status on student scientific achievement. Their results indicated that socioeconomic status has a significant and substantial impact on individual scientific achievement. Relatedly, Tsai, Yang, and Chang [1] reported that parental education level is the one independent latent variable that has direct effects on the science performance of eighth-grade Taiwanese students. In the same study, those authors further reported the finding that PEL had indirect but considerable effects on the science performance of students via its effects on the attitudes of students toward science. That said, no relevant research has pointed out the relationship between the socioeconomic status and ocean literacy of students, in spite of the fact that marine education has always been an important part of science education. Therefore, it is necessary to explore the relationship between PEL and ocean literacy by using PEL as the indicator of socioeconomic status.

### 2.3. Attitude and Ocean Literacy

The attitudes of students constitute one of the most important factors affecting their academic achievement. Many studies have shown that attitude has a positive impact on academic achievement. Positive learning attitudes and high levels of motivation for learning are very helpful for students’ learning and can effectively improve students’ academic achievement. Student attitude toward the environment or ocean is also well-documented to be a factor that influences ocean literacy [16,17]. At the same time, there is no relevant literature directly indicating the relationship between students ATO and ocean literacy. However, a study by Greely [16] did investigate the mediating role of attitudes about the ocean in determining ocean literacy. The results of that study revealed that content knowledge and environmental attitudes significantly contributed to ocean literacy. In an earlier study, Fortner and Mayer [17] conducted a baseline investigation of the knowledge and attitudes of students regarding the ocean and Great Lakes. They found that the participating students exhibited limited knowledge in general, with only 37.6% and 48.3% of the questions they posed being answered correctly by fifth graders and ninth graders, respectively. Their results further showed a relationship between student attitudes toward the ocean and Great Lakes and their knowledge of the same, with more positive attitudes being exhibited by those with high levels of knowledge. More recently, a study on ocean literacy by Cudaback provided a summary of the affective factors that should be considered in projects aimed at promoting ocean literacy, as well as a summary of the ocean-related topics that college students are interested in [18].

## 3. Materials and Methodology

### 3.1. Ocean Literacy and Contextual Factors (OLCF) Model

Based on the literature review above, the measurement model and theoretical model of ocean literacy and contextual factors (OLCF) is proposed in Figure 1 and Figure 2. In the proposed model of OLCF, both ATO and ocean literacy are predicted by PEL, while ocean literacy is also predicted by ATO itself. Empirical evidence that the development of ATO is strongly influenced by parents was previously provided by George and Kaplan [19]. Thus, we suspect that PEL has implications for ATO. Meanwhile, students’ performance has likewise been shown to be influenced by PEL [10]. For this reason, PEL is depicted as having a direct effect on ocean literacy in this model. Furthermore, the aforementioned study by Greely found that ocean literacy is significantly contributed to by both environmental attitudes and content knowledge [16]. Therefore, we also sought in this study to investigate how ocean literacy is affected by ATO.

### 3.2. Measures

#### 3.2.1. Ocean literacy

The ocean literacy inventory consisting of 48 items with good psychometric qualities (e.g., reliability, validity, unidimensionality, and differential item functioning) developed by Tsai and Chang [20] was used to measure student ocean literacy. The seven basic principles of ocean literacy published by the NMEA, as well as ocean literacy and knowledge in general, served as the basis for this inventory [21]. The full test consists of seven subscales: (1) Features of the ocean, (2) the ocean and its life shape earth, (3) weather and climate, (4) the ocean made earth habitable, (5) the diversity of life and ecosystems, (6) the ocean and humans are interconnected, and (7) the ocean is largely unexplored. A total of ten multiple-choice items and 38 single-choice items are included in the questionnaire. Each correct answer has a value of 1, whereas a value of 0 is recorded for any wrong answers. To see the content of the completed items, please refer to the relevant study by Tsai and Chang [20].

#### 3.2.2. Parental Education Level

As suggested by both Myrberg and Rosén [8] and Tsai, Yang, and Chang [1], the present study used paternal and maternal education levels to determine the index for PEL, with a four-point Likert scale being used to designate the PEL for each participating student (1 = junior high school or below; 2 = completed senior high school degree; 3 = completed university degree; 4 = completed master’s degree or above).

#### 3.2.3. Attitude toward the Ocean

The index for ATO in this study was based on three statements: ‘I enjoy learning marine knowledge’; ‘marine science is boring’; and ‘I like marine science’. The TIMSS international survey used three items (‘I enjoy learning science’; ‘science is boring’; and ‘I like science’) to define science attitude and explore the sources of variability in science performance [22]. Similarly, ATO was obtained in this study by asking the students to indicate their agreement or lack thereof with the three revised items mentioned above. A four-point Likert scale was used to provide the response options for each item (1 = disagree strongly; 2 = disagree somewhat; 3 = agree strongly; 4 = agree somewhat). The second item was reverse scored. A higher score was indicative of a more positive ATO for the given respondent.

### 3.3. Study Process and Participants

The sample of participants in this study consisted of senior high students (aged 16–18 years old) in Taiwan, with three stages of stratified random sampling being performed. In the first stage, schools from the southern, northern, eastern, and central regions of Taiwan were subjected to stratified random sampling. In the second stage, two classes from each of the schools chosen in the initial step were randomly selected. Ultimately, a total of 1050 students from 40 classes at the various schools were selected to complete the actual test. The test was a 50-min, paper-and-pencil test that allowed the students to respond with open-ended answers. The test was administered from May to June 2017 by test administration committee members who had all undergone standardized testing training before visiting each of the schools to administer the test. After the administration of the test had been completed, the accuracy of the test data was verified through compilation and double-checking of the results. A total of 1050 students were asked to fill in the questionnaire. After the self-reported questionnaire was administered to the students, the final total of 945 valid respondents was found to consist of 549 (58.1%) male students and 396 (41.9%) female students. Of the students who ultimately composed the valid sample, students from the northern, southern, and central regions of Taiwan accounted for 12.3%, 60.7%, and 27.0% of the sample, respectively.

### 3.4. Data Analysis

In order to achieve the purpose of the research, the relationships among PEL, ATO, and ocean literacy were explored following the establishment of measurement invariance across genders. A series of advanced analytical methods including confirmatory factor analysis (CFA) and multiple-group structural equation modeling (MGSEM) was carried out by using Mpls 8 [23] to achieve the analytical objectives.

The calculation of various descriptive statistics, including the kurtosis, mean, and skewness for each of the observed variables, was performed, and the identification of those latent variables that could be estimated based on the observed indicators was then performed by using the CFA model. The goodness-of-fit indices were used to assess the analysis results of the CFA model. If a good fit was obtained for the indices of the CFA model, then the hypothetical latent variables were considered to be reliable and appropriate [1]. The Tucker–Lewis index (TLI), normed fit index (NFI), comparative fit index (CFI), root mean squared error of approximation (RMSEA) index, and standardized root mean square residual (SRMR) index recommended by Beauducel and Wittmann [24], Fan and Sivo [25], and Hu and Bentler [26] were used to assess the CFA model-data fit.

The measurement invariance across genders of the hypothesized OLCF model was established via MGSEM comparisons. A hierarchical examination of seven nested models with more and more parameter constraints was performed, with those seven models being configural invariance, invariance of intercepts of measured variables, invariance of factor loadings of measured variables, invariance of structure covariance, invariance of residuals variance of measured variables, invariance of intercepts of latent variables, and invariance of disturbances of latent variables. If the model could still yield a good fit even with an increasing number of constrained parameters, it would provide us with greater and greater confidence in the stability and validity of the hypothesized OLCF model. That is, the invariance in the associations among the latent and observed variables in the model would be frequently reflected by the measurement invariance of the hypothesized model across genders.

The ΔCFI and ΔRMSEA suggested by Cheung and Rensvold [27] and Meade, Johnson, and Braddy [28] were used to assess the restricted model that was tested hierarchically in the current study. As such, the null hypothesis of invariance was not rejected in the event that the change value of ΔRMSEA between two nested models was equal to or smaller than 0.007 or that of ΔCFI was equal to or smaller than 0.01 [27,28].

## 4. Results

### 4.1. Descriptive Statistics of all Observed Variables

Table 1 presents descriptive statistics, including the mean values, standard deviation values, skewness values, and kurtosis values, for all of the observed variables. The skewness values ranged from −1.362 to 0.31, and the kurtosis values ranged from −1.093 to 2.327. The absolute values of kurtosis and skewness were all lower than the cut off values of 3 and 8 recommended by Kline [29] and by Tsai, Yang, and Chang [1], respectively. Furthermore, all the data were, according to the preliminary data analysis, within acceptable ranges of the normal distribution. As such, this study utilized the maximum likelihood (ML) estimation method.

### 4.2. Reliability and Validity of Measurement Model and OLCF Model

The measurement model is presented in Figure 1. In order to estimate the validity and reliability of the model, as well as to determine which of the hypothetical latent variables were reflected in two or more of the observed variables, a CFA was conducted. According to the results, the proposed CFA model had a good fit with the overall sample.

The goodness-of-fit statistics and indices were as follows: χ^2^ = 136.38; *df =* 51; *p* < 0.001; CFI = 0.980; TLI = 0.974; NFI = 0.969; RMSEA = 0.042; and SRMR = 0.034. Statistical significance at the 0.001 level was achieved for all of the standardized factor loadings of the observed variables in the measurement model. The composite reliabilities (CR) were 0.83, 0.83, and 0.85 for PEL, ATO, and ocean literacy, respectively. The values were all greater than 0.60, as recommend by Fornell and Larcker [30]. The average variance extracted (AVE) values were 0.72, 0.65, and 0.46 for PEL, ATO, and ocean literacy, respectively. They were all greater than 0.25, as recommended by Hair, Black, Babin, Anderson, and Tatham [31]. According to these results, the latent variables in this study were all somewhat valid and reliable.

The hypothesized model (OLCF model) that was proposed in the current study is shown in Figure 2; PEL was the independent variable which was found to have causal effects on students’ ocean literacy. In addition, the ATO variable was also found to mediate the relationship between PEL and ocean literacy.

The MGSEM analysis was carried out to determine whether the relationships of the latent variables in the hypothesized model fit the data separately for both gender groups. According to the results, the hypothesized model was identified and found to fit the data adequately. For the male group, the goodness-of-fit indices were CFI = 0.981, TLI = 0.975, NFI = 0.963, RMSEA = 0.044, and SRMR = 0.035, while those for the female group were CFI = 0.960, TLI = 0.948, NFI = 0.929, RMSEA = 0.055, and SRMR = 0.053. With the exception of PEL to ATO for the male group, statistically significant results (*p* < 0.05) were obtained for all of the coefficients shown in Figure 2 for the two groups. Overall, these results demonstrated both that the hypothesized model was potentially plausible and that it fit across the two gender groups.

### 4.3. Measurement Invariance of the OLCF Model

Using a series of nested models in which successive equivalence constraints in the model parameters across groups were established, the measurement invariance of the hypothesized model was tested in a hierarchical manner in order to determine whether the model had validity and statistical equivalence across the gender groups [32]. The first level model, or baseline model, was the configural model (Model 1), which had the fewest restrictions put on the parameters. The specifications for this model included the specification that each group had a similar structure, as well as the specification that each group had the same pattern of fixed and freely estimated parameters. The second level model (Model 2) also included factor loading invariance, or metric invariance. At this level, testing of a model in which the factor loadings across the groups had equality constraints placed on them was conducted. For the third level model (Model 3), additional constraints were imposed on the intercepts of the measured variables in order to test scalar invariance.

In Model 4, the fourth level model, equality constraints placed on the intercepts of the latent variables were added to Model 3. When the intercepts and factor loadings of both the latent and measured variables are all constrained such that they are equal across groups, it suggests that an identical measurement unit and structure pattern are present in the groups. Model 5 was the same as Model 4, except that equality constraints of structure covariance were also included in Model 5. The next level model (Model 6) was established by the imposition of constraints on the factor unique variance of the latent variables, which allowed for the testing of the invariance of disturbances across groups. Lastly, the final and most restricted model, Model 7, was the same as Model 6, except that residual equality constraints of measured variables were also included in Model 7, and the resulting model was tested for residual variance of the measured variables across groups [32,33].

The hypothesized models for males and females were analyzed using multi-group analyses, the results of which are shown in Table 2. For all the models (Model 1–Model 7), the goodness-of-fit RMSEA values were all less than 0.05, with the maximum value being 0.047, which indicated that the model adequately fit the data. The SRMR indices were all less than the critical value 0.08 [26], and this also showed that all the models fit the data. Regarding the detection of measurement invariance, in the first step, the ΔCFI and ΔRMSEA values between Model 2 (metric invariance) and Model 1 (configural invariance) were less than 0.01 and 0.007, respectively. These results indicated that, for both groups, acceptable invariance of the factor loadings of the measured variables was found. Meanwhile, in Model 3, equal settings were imposed for the intercepts and factor loadings of the measured variables for both groups. The ΔCFI was less than 0.01 and ΔRMSEA was less than 0.007, meaning, in other words, that gender did not cause any variation in the intercepts of the measured variables. Next, the estimation of Model 4 was performed with equality constraints placed on the intercepts and factor loadings of both the latent and measured variables. Compared with Model 3, the ΔCFI and ΔRMSEA indices all fit the criteria, which indicated that the intercepts of the latent variables were invariate across the genders. Fifthly, Model 5 was estimated with equality constraints on the invariance of structure covariance. Compared with Model 4, the ΔCFI was less than 0.01 and the ΔRMSEA less than 0.007, which indicated that the structure covariance did not vary across the genders. Subsequently, in Model 6, equality constraints were placed on disturbances of the latent variables across the groups. The results also revealed that invariance of disturbances of latent variables was holed. Finally, in Model 7, the residual variance of the measured variables were constrained as equal across the groups. The ΔCFI and ΔRMSEA were still less than 0.01 and 0.007, respectively. These results demonstrated support for and the tenability of invariance in the residual variance of the measured variables across genders. To summarize, the multiple-group analysis results supported the conclusion of measurement invariance across the participating male and female students aged 16–18 years old.

### 4.4. Effects of PEL and ATO on Ocean Literacy

Table 3 presents the indirect, direct, and total effects of the various latent variables on ocean literacy under the condition of measurement invariance across the genders. According to the analysis conducted with the latent variables, the ATO was found to mediate the effect of PEL on ocean literacy. The total effect of PEL on the students’ ocean literacy varied between the genders. The estimated influence for the female students (0.270) was larger than the estimated influence for the male students (0.172). For the female group, there was a considerable effect of parental education that was indirectly related to OL through its effect on attitude toward the ocean (coefficient = 0.16, *p* < 0.05).

In contrast, the effect of PEL that was not indirectly related to ocean literacy through its effect on ATO (coefficient = 0.01, *p* > 0.05). For the effects of PEL that were directly related to ocean literacy, the coefficients were 0.241, 0.171, and 0.191 for the female students, male students, and total sample, respectively. Similarly, for the effects of ATO that were also directly related to ocean literacy, the coefficients were 0.179, 0.091, and 0.120 for the female students, male students, and total sample, respectively. However, the direct effect of PEL was stronger than that of ATO. Overall, substantial effects of the students’ backgrounds on their ocean literacy were exhibited in the two groups, and the results of the MGSEM analyses supported the measurement invariance of the hypothesized model across genders.

## 5. Discussion

This study aimed to construct a SEM model to analyze the effects of parental educational level on student ocean literacy through attitude toward the ocean based on gender invariance in a representative sample of Taiwanese students aged 16–18 years old. In this study, the latent variable of student attitude toward the ocean was measured by three items that were modified from the TIMSS questionnaire scale. The latent variable of parental educational level was measured by the two observed variables of paternal education level and maternal educational level. Student ocean literacy scores were taken from the students’ responses to the ocean literacy questionnaire that was developed by Tsai and Chang [20]. In this study, a structural equation model of the relationships between parental educational level and attitude toward the ocean and their effects on student ocean literacy was proposed and evaluated to determine how these two factors influence students’ ocean literacy based on gender invariance. The multiple-group analysis results supported the conclusion of measurement invariance across the genders, in addition to indicating that the model fit the data for both the male and female students well. Through the estimations of the indirect, direct, and total effects on ocean literacy of parental educational level, the relationship between parental educational and ocean literacy was greatly clarified.

The correlation coefficient values among all the variables indicated that parental educational level and attitude toward the ocean were both associated with ocean literacy after the establishment of gender invariance. Also, higher parental education level was also found to be positively related to ocean literacy, with students whose parents had higher educational levels scoring higher in ocean literacy than those whose parents had lower educational levels. Parental education level contributed to ocean literacy both directly and indirectly through its effects on attitude toward the ocean in the female students. These findings were consistent with those of previous studies [7,9,34,35]. For instance, Kalender and Berberoglu [7] indicated that parental educational level is the most important variable of all the variables in terms of its effects on student academic achievement. Campbell et al. [9] also demonstrated that students who have higher reported parental education levels tend to have higher assessment scores. The current study also found that there was a considerable indirect effect of parental educational level on ocean literacy that was mediated through attitude toward the ocean in the female student group but not in the male student group. However, the influence of parental educational level on attitude toward the ocean was not statistically significant. This study also found that parental educational level only has a direct effect on ocean literacy for male students, whereas it does not have an indirect effect on ocean literacy for male students. Moreover, parental educational level contributed to ocean literacy through its effects on attitude toward the ocean, although the indirect effect for the male group was weaker than that for the female group. Therefore, in the female group, it appears that students whose parents have higher education levels tend to enjoy the ocean and enjoy learning marine knowledge more than students with parents of lower educational levels. Female students are also more likely to have greater ocean literacy when their parents have higher levels of education.

Students in Taiwan do not officially start their study of practical marine science, which is included in the overall earth science curriculum, until the ninth grade, so the students in this study had only been exposed to formal marine science education for a short period. Relatedly, when students are relatively young, parents are most likely to primarily teach their children how to perform well in the classroom, as well as to encourage them to enjoy learning and develop a positive attitude and interest in science. As such, if the study of marine science education is linked to specific goals (e.g., getting a good job), it may be accompanied by an authoritarian parenting style. It will also be even more difficult to cultivate students’ positive attitude toward learning. In Taiwan, students aged 16–18 years old typically have relatively good opportunities to attend university before they begin their working careers, so students may not immediately feel that they need any knowledge of science in general or marine science in particular to help them achieve the goal of attending university. As such, it is recommended that parents try to encourage the development of scientific interests and positive attitudes toward marine science in their children. Through such encouragement, the children will, as their academic performance progresses, naturally begin to understand that science or marine science education can help them gain acceptance to a better university, develop their internal interests, and obtain better employment opportunities. These findings regarding marine science education are consistent with findings regarding other fields of study (e.g., other science fields and mathematics) [1,7,9,15,34,35]. Parental educational level is the most important variable of all the variables in terms of its effects on student academic achievement. Therefore, according to the results of this study and previous studies, it is necessary to encourage parents to make additional efforts to enhance their children’s positive attitudes toward learning (e.g., toward science learning in general and marine science learning in particular). Moreover, while the educational level of students’ parents typically does not change, the learning attitudes of students can be cultivated. Therefore, as students engage in learning, their teachers, parents, and other family members should try to help enhance their learning attitudes. Even the relevant systems within schools should be coordinated to enhance students’ attitudes towards learning and thus improve their learning achievement.

While the present study specifically found both attitude toward the ocean and parental education level to be important influences on the ocean literacy of students, it is also possible that other related variables play important roles in such literacy, and as such, those other variables also deserve further study. Furthermore, the questionnaire employed in the current study has some limitations. Additional research is thus still needed, and it is recommended that future studies measure more specific variables that may affect students’ ocean literacy, such as self-confidence, classroom pedagogical style, and practice and experiences involving ocean activities. More complete and precise data collection should allow researchers to make better inferences, and thus enhance our understanding of the relationships among these variables. In any event, the focus of the present study was restricted to student personality and socioeconomic status factors, so it is not possible for this study to explain how other background factors affect ocean literacy. In addition, the data collection for this study was conducted solely in Taiwan, in spite of the fact that the questionnaire used in the study is available in various languages. Therefore, cross-international data collection should be performed in future research, as data from other countries should help to clarify any differences in ocean literacy, as well any differences in the factors determining ocean literacy, among the students of various nations. Relatedly, the subjects investigated in the current study should remain very important subjects for future research.

## 6. Conclusions

This study sought to gain a more comprehensive understanding of how the factors of parental education level and student attitude toward the ocean influence the ocean literacy of students in Taiwan after establishing measurement invariance across genders. Higher parental education level was found to be positively related to ocean literacy. Parental education level contributed to ocean literacy both directly and indirectly through its effects on attitude toward the ocean in the female students. Therefore, in the female group, it appears that students whose parents have higher education levels tend to enjoy the ocean and enjoy learning marine knowledge more than students with parents of lower educational levels. Female students are also more likely to have greater ocean literacy when their parents have higher levels of education. Finally, although some of the results of this study are similar to those of previous research regarding PEL, ATO, and science achievement, there was no previous relevant research on the relationships among PEL, ATO, and OL. The results of this study therefore serve to clarify the relationships among these three variables. The parental education level plays a very important role in the students’ learning achievement (e.g., in science, mathematics, or marine education).

## Figures and Tables

**Figure 1 ijerph-16-03672-f001:**
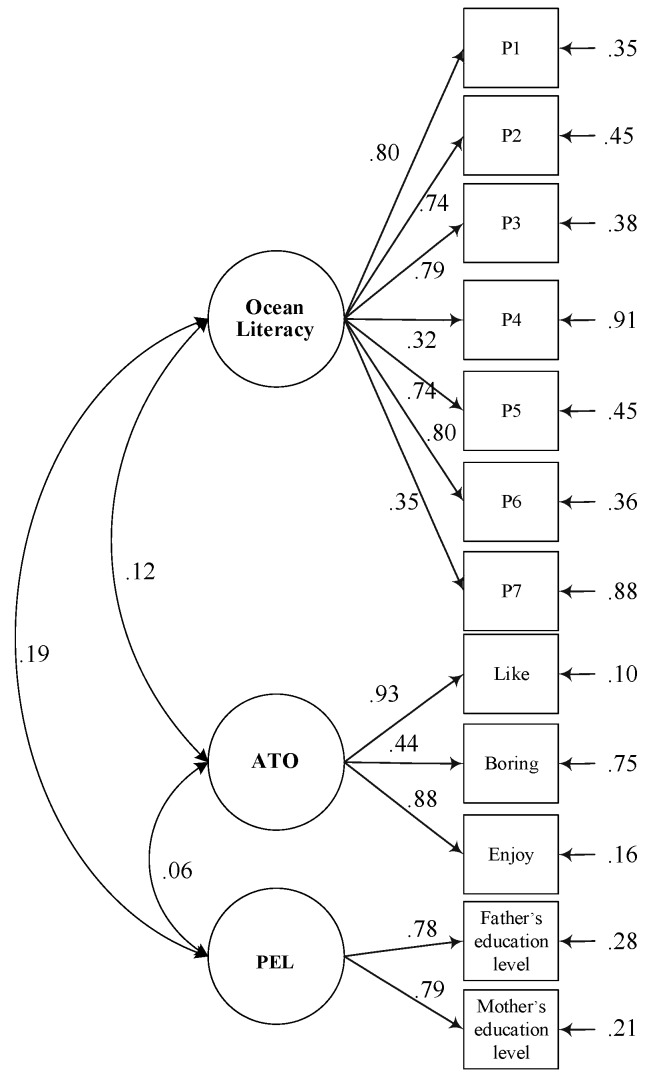
Measurement model of latent variables. Note: *N* = 945; PEL = parental educational level; ATO = attitude toward the ocean. All values are statically significant (*p* < 0.001).

**Figure 2 ijerph-16-03672-f002:**
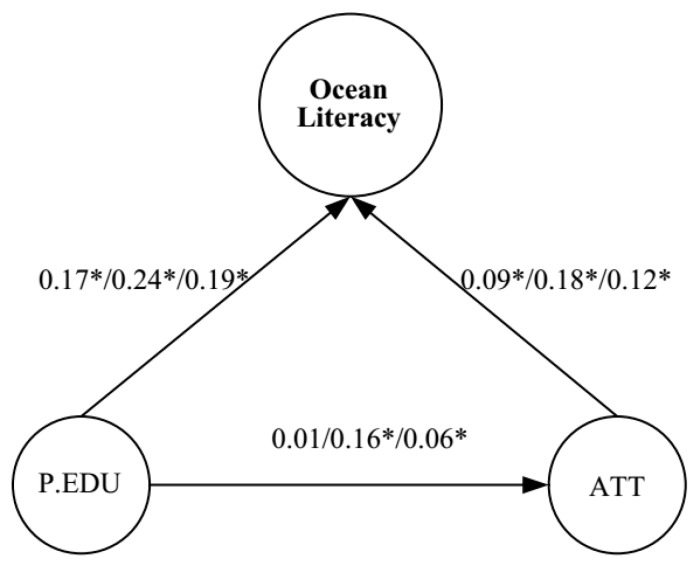
Standardized estimates of relations and effect sizes in structural model. Note: Female group *N* = 396, male group *N* = 549, and total sample *N* = 945. PEL = parental educational level; ATO = attitude toward the ocean; the first value, second value, and third value shown are for the male group, the female group, and the total sample, respectively. * indicates statistically significant (*p* < 0.05).

**Table 1 ijerph-16-03672-t001:** Descriptive statistics of all observed variables.

	Total Sample				Male				Female			
Item/Observed Variable	M	SD	S	K	M	SD	S	K	M	SD	S	K
1. PEL_Paternal education level	2.70	0.84	−0.09	−0.60	2.71	0.84	−0.17	−0.48	2.68	0.85	0.02	−0.74
2. EL_Maternal education level	2.62	0.75	−0.14	−0.05	2.62	0.77	−0.22	0.09	2.63	0.73	−0.03	−0.30
3. ATO_Enjoy	3.32	0.86	−0.15	0.76	3.27	0.90	−0.19	0.75	3.37	0.79	−0.01	0.59
4. ATO_Boring	3.50	0.97	−0.34	0.11	3.38	1.03	−0.23	−0.01	3.67	0.87	−0.37	0.22
5. ATO_Like	3.29	0.85	−0.15	0.91	3.23	0.89	−0.15	0.86	3.37	0.78	−0.37	0.81
P1: Features of the ocean	17.46	3.39	−1.36	2.09	17.43	3.73	−1.36	1.65	17.50	2.86	−1.21	2.33
P2: The ocean and its life shape earth	6.21	1.90	−0.53	−0.27	6.11	2.04	−0.49	−0.56	6.34	1.67	−0.46	0.08
P3: Weather and climate	8.89	2.41	−1.09	0.99	8.81	2.65	−1.02	0.46	9.20	2.02	−1.01	1.52
P4: The ocean made earth habitable	1.24	0.66	0.48	0.48	1.22	0.66	0.32	0.25	1.27	0.66	0.73	0.77
P5: The diversity of life and ecosystems	7.34	2.26	−0.65	−0.15	7.43	2.45	−0.73	−0.30	7.21	1.95	−0.52	0.09
P6: The ocean and humans are interconnected	9.92	2.71	−1.01	0.53	9.76	3.02	−0.91	−0.09	10.13	2.20	−0.99	1.50
P7: The ocean is largely unexplored	0.95	0.73	0.08	−1.09	1.01	0.75	−0.01	−1.23	0.87	0.68	0.16	−0.83

Note: PEL = parental educational level; ATO = attitude toward the ocean; M = mean; SD = standard deviation; S = skewness; K = kurtosis.

**Table 2 ijerph-16-03672-t002:** Fit indices for multi-group analysis across genders.

Model	χ^2^	*df*	CFI	RMSEA	SRMR	Model Comparison	ΔCFI	ΔRMSEA
Model 1 Configural invariance	216.794	102	0.973	0.035	0.0357	-	-	-
Model 2 Invariance of factor loadings of measured variables	233.874	111	0.971	0.034	0.0370	2 vs. 1	0.002	0.001
Model 3 Invariance of intercepts of measured variables	297.555	123	0.969	0.039	0.0371	3 vs. 2	0.002	0.005
Model 4 Invariance of intercepts of latent variables	301.797	126	0.969	0.039	0.0404	4 vs. 3	0.000	0.000
Model 5 Invariance of structure covariance	302.414	127	0.969	0.038	0.0404	5 vs. 4	0.000	0.001
Model 6 Invariance of disturbances of latent variables	371.111	129	0.963	0.044	0.0508	6 vs. 5	0.006	0.006
Model 7 Invariance of residuals variance of measured variables	433.1	141	0.954	0.047	0.0483	7 vs. 6	0.009	0.003

Note: Female group *N* = 396, male group *N* = 549. df = degree of freedom; CFI = comparative fit index; RMSEA = root mean squared error of approximation; SRMR = standardized root mean square residuals.

**Table 3 ijerph-16-03672-t003:** Direct, indirect, and total effects of latent variables on ocean literacy.

Effects	Paths	Female	Male	Total Sample
Direct	PEL → ATO	0.164	0.002	0.061
	PEL → ocean literacy	0.241	0.171	0.191
	ATO → ocean literacy	0.179	0.091	0.120
Indirect	PEL → ocean literacy	0.029	0.001	0.007
Total	PEL →ATO	0.164	0.002	0.061
	PEL → ocean literacy	0.270	0.172	0.198
	ATO →ocean literacy	0.179	0.091	0.120

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
