# Peer review of "An Assessment of Factors Related to Ocean Literacy Based on Gender-Invariance Measurement"

_ijerph, 2019, doi:10.3390/ijerph16193672_

Round 1

Reviewer 1 Report

The manuscript is of potential interest to the readership of this journal, but there are major issues that must be addressed before the article could be published.

1/ * The literature review should be more carefully synthesised and structured. The use of sub-headings and signposting would help the reader to follow the argument being developed through the paper.

2/ * There does not appear to be an explicit theoretical framework - this makes it difficult to explain the differences between outcomes of the research groups. Currently the manuscript appears to be somewhat descriptive and a theoretical.

3/ * The results section requires far greater organisation and structuring. The analysis is too general, and the reported results are somewhat selective. This section needs to be more carefully and systematically constructed.

4/ * Further, the analysis and findings must be critical and interpretive rather than just descriptive.

5/ * The final discussion and conclusion should make it clear how the findings contribute to new knowledge.

6/* The Methodology lacked suitable detail.

7/* Methodology of the central work is exposed appropriately.

8/* Research methodology based on research goals is poor on absent.

9/* More recent bibliography is necessary. Furthermore, the reference list is a little bit weak. Before I can make a final decision on the paper, please refer to more references and upload a new version. It is suggested that the author(s) can consider the following paper related to parental education students' scientific achievement to strengthen the background and conclusions of the study:

Papadakis, S., Kalogiannakis, M., & Zaranis, N. (2019). Parental involvement and attitudes towards young Greek children’s mobile usage. International Journal of Child-Computer Interaction. doi: https://doi.org/10.1016/j.ijcci.2019.100144.

10/* The academic writing needs work.

11/* The discussion should be more concise and the outcomes should be discussed in relation to the existing research. The language style of this section has to improve.

12/* Recommendations should also be given for practice and further research.

13/* Some references within the paper body are not correct.

14/* The paper must be correctly formatted according the journal format

15/* Similarity check with iThenticate revealed a similarity index of 33% which is considered TOO high. A maximum of around 60 quoted words is accepted per paper. There are 8 papers with over 60 words. No previously copyrighted material can be used.

18/* In preparing a revised manuscript, please also include a table of how you have responded to each of the issues listed above point by point.

 I look forward to receiving your revised manuscript in the near future.

Reviewer 2 Report

this was a very extensive study. sound research design,method, and extensive description of results with diagrams and grafts.  I look also for antidotal notes and this was provided along with the statistical results. the Significance of the conclusion for future was informative. I found the student of ocean literacy interesting for students in Taiwan and after reading the article to see the significance.  The connection of parental education to student attitude and the study of ocean marine science was sound.

For suggestions I would ask that they expand in the conclusion for application to other studies on parental education and student attitude toward learning to expanded other areas. The study was limited to Ocean Literacy and marine sciences in Taiwan. So this would be accepted with minor revision.

Round 2

Reviewer 1 Report

This is a well written paper and I thoroughly enjoyed reviewing it:

The paper contains new and significant information adequate to justify publication. The paper demonstrates an adequate understanding of the relevant literature in the field and cite an appropriate range of literature sources. The results are presented clearly and analyzed appropriately. The paper clearly expresses its case, measured against the technical language of the field and the expected knowledge of the journal's readership. The paper has attention been paid to the clarity of expression and readability, such as sentence structure, jargon use, acronyms, etc.